# Bridging the Gap between the Pressing Need for Family Skills Programmes in Humanitarian Settings and Implementation 

**DOI:** 10.3390/ijerph19042181

**Published:** 2022-02-15

**Authors:** Aala El-Khani, Rachel Calam, Karin Haar, Wadih Maalouf

**Affiliations:** 1Prevention, Treatment and Rehabilitation Section, Drug Prevention and Health Branch, United Nations Office on Drugs and Crime (UNODC), Division of Operations, Wagramer Strasse 5, A-1400 Vienna, Austria; karin.haar@un.org (K.H.); wadih.maalouf@un.org (W.M.); 2Division of Psychology & Mental Health, University of Manchester, Manchester M13 9WL, UK; rachel.calam@manchester.ac.uk

**Keywords:** refugee, family skills, parenting, displaced population, war, conflict

## Abstract

A supportive environment with nurturing caregivers is essential for the healthy development of children. For children who have been exposed to extreme stress, such as humanitarian contexts, the need for strong, healthy, nurturing caregiver relationships may assume even greater importance. Much research has been building to position family skills interventions as a key tool in encouraging safe and supporting relationships between caregivers and children, thus preventing many problem behaviours and poor mental health. While there is substantial evidence of the effectiveness of family skills interventions in high-income and stable contexts, evidence of interventions that have been tested in humanitarian and challenging settings, such as contexts of refugee and displacement, are far fewer. Despite the role that family skills interventions can play in protecting children from current and future challenges, there is a significant lack of such interventions being utilised in humanitarian settings. We put forward seven likely reasons for this lack of uptake. Furthermore, the Strong Families programme, a UNODC family skills intervention, is presented as an example of an intervention that aims to bridge this gap of interventions that meet the need for humanitarian and contexts of extreme stress. More research is needed to unpack the content, delivery mechanisms and reach of family skills programmes to further aid programme developers in investing in efforts that might provide significant sustained impact for families in humanitarian contexts.

## 1. Introduction

For children, living through the stress of conflict, displacement or circumstances of inequality will often mean facing emotional, physical and cognitive challenges [1,2]. Much research now indicates these challenges affect not only how children develop physically and mentally but also how they interact with their family members and society. Experiencing extended stress is a strong predictor of negative social and health outcomes, including partaking in risky behaviours such as drug use, low school attainment, delinquency, violence, and poor mental health [3,4]. Further, the consequences of extensive stress have also been found to lead to the intergenerational transmission of trauma to future generations [5]. With an estimated one in five children in the world now living in areas affected by armed conflict [6], interventions to buffer the effects of prolonged stress on children are urgently needed [2].

A supportive environment with nurturing caregivers is essential for the healthy development of children [7,8], and this has been established in diverse cultural and social contexts [9]. Positive family interactions can serve as a protective factor, supporting children’s development of resilience and providing them with a sense of hope for the future as families face the upheaval of migration or living in a low resource setting [10,11]. Studies exploring the links between war and parenting in ongoing humanitarian crises highlighted the key role that parents or primary caregivers play [7]. However, parenting in contexts of high stress reduces the likelihood of the parent’s ability to provide children with positive interactions that promote healthy psychosocial and physical development; instead, it increases the likelihood of engaging in harsh, inconsistent parenting and other negative coping mechanisms, which increases children’s risk of enduring emotional and behavioural problems [12,13].

Over the past two decades, much research has been building to position family skills interventions as a key tool in encouraging safe and supporting relationships between caregivers and children, thus preventing many problems and risky behaviours and poor mental health [14]. Family skills training programmes are designed to strengthen protective factors and reduce risk factors within the family [15]. Family skills programmes offer a combination of parenting knowledge, skill-building, competency enhancement and support [16]. They aim to strengthen family protective factors such as communication, trust, problem-solving skills and conflict resolution and strengthen the bonding and attachment between caregivers and children.

The UNODC WHO International Standards on Drug Use Prevention [17], INSPIRE initiative to end violence against children [18], WHO-led Violence Prevention Alliance in their guidance to prevent youth-based violence [19,20], and WHO/UNICEF Helping Adolescents Thrive initiative to prevent and promote mental health in adolescence [21] have all recommended evidence-based family or parenting skills programmes as a common denominator intervention, serving multi-outcome initiatives.

## 2. Barriers to Implementation

While there is substantial evidence of the effectiveness of parenting and family skills interventions in high-income and stable contexts, interventions that have been tested in low and middle-income countries, and likewise in the contexts of refugee and displacement, are far fewer [2]. However, there are four recent exceptions of studies indicating the effectiveness of parenting and family skills interventions in refugee and displacement contexts in reducing child maltreatment through utilising nonphysical consequences to non-desirable behaviour and more positive parenting approaches, as well as improving parental and child mental health, such as reduced rates of depression [9,22,23,24]. For children living under extreme stress, the need for strong, healthy and nurturing caregiver relationships may assume even greater importance [25] when other extended support systems in their lives, such as community and extended family, may be compromised. Despite the key role family skills interventions can play in protecting children from current and future challenges, there is a significant lack of family skills interventions being utilised in humanitarian settings. We put forward seven likely reasons for this lack of uptake:

### 2.1. Competing Priorities

There are competing priorities and needs for such populations where food and shelter and other immediate social and medical services, when resources are availed, take priority. The relative rarity of parenting family skills packages parallels levels of provision of any other packages of psychosocial support and, in many instances, physical and mental health services. Mental health and family functioning are often deemed a secondary need, with funding not allocated in such contexts as a priority. The authors’ experience with working with families in a humanitarian context is that they are often in ‘fight or flight’ mode, and they themselves do not seem to be prioritising their parenting. When allowed to explore their needs, caregivers describe much anxiety, worry, and sadness at their feelings on not knowing how to support their children through the emotional and behavioural changes they witness [26]. They are also keen to gather information as soon as basic needs are met [27].

### 2.2. Programme Intensity

Existing family skills programmes are often high intensity and multi-session, which assume a context of families being stable in one area or able to make themselves available for the length of the programme, which in some cases may last up to 20 weeks. This can be impractical for humanitarian contexts, where families may only be in one place for a short period of time. Moreover, caregivers may be busy prioritising obtaining their essential daily needs, such as water, food and sanitation. Recently, there has been recognition for the need for a movement towards developing brief interventions. There is growing recognition that a range of flexible, low-intensity interventions is required to extend the reach and impact of family skills support [28]. A welcome addition to the field has been a systematic review of the evidence for the efficacy and effectiveness of brief parenting interventions, defined as less than eight sessions in duration, in reducing child externalising behaviours [29]. The review identified nine papers summarising the results of eight studies with 836 families in five countries that met the inclusion criteria. Most of these studies were conducted in the last few years, despite the literature search spanning more than a 20 year period. All studies were conducted in high-income countries except one conducted in Panama [30]. Furthermore, seven of the studies utilised the same parenting intervention.

### 2.3. Quality of Evidence

Much psychosocial work involving children and their caregivers may be being carried out, but without randomised controlled trial (RCT) frameworks and well-designed research, they may not be identified by academic reviews. Even the limited RCTs existing do not avail information on feasibility and ease of scale-up and reach. In addition, in terms of measurability, it is far easier to collate information for donors, such as the number of items delivered to beneficiaries, than it is to measure mental health, even though there are standardised ways of measuring mental health and well-being in terms of reach and impact [27]. This is likely due to poor research infrastructures in place (if any), coupled with limited resources and competing interests of where to earmark funds and the associated costs of research assistants for data collection and the costs of utilising some data collection tools.

### 2.4. Awareness

There has also been a lack of awareness and knowledge among policymakers and field implementation staff of the broad and significant benefits of family skills interventions in such contexts. This is shifting considerably, and over the past decade, the family skills training field has seen a significant rise in recognition from the international community. This is likely due to both the steep rise in displacement globally as well as more funding being allocated to develop research infrastructures that are monitoring and evaluating the uptake and utilisation of such programmes in humanitarian contexts.

### 2.5. Programme Design

Very few family skills programmes have been designed to serve the needs of families living under extreme stress, such as in humanitarian settings [2]. In a recent review of existing evidence of parenting programmes in low and middle-income countries (LMIC) [2], only one evaluated intervention incorporated sessions tailored to a conflict-affected population in Northern Uganda [31]. Without being contextually and culturally relevant, families may be less likely to engage.

### 2.6. Infrastructure

Family skills programmes often require a well-resourced infrastructure which also includes high royalty fees to developing institutions common in high-income countries, costly programme materials (such as DVD’s, cameras, etc.) and requirements for facilitators to have psychology degrees or similar specialised backgrounds. All of these elements pose a significant challenge to humanitarian programming either to test such interventions in contexts where such profiles of facilitators exist or more valuably to move them to scale post-pilots.

### 2.7. Training Requirements

Family skills training often requires intensive face to face training, for which programme master trainers travel to the proposed implementing country. This is often due to the complexity and or intensity of the intended intervention imposing such a requirement to ensure fidelity of its implementation. This can prove as an impediment to reaching volatile or hard to reach areas with training and information in countries where conflict is active or where safety for non-nationals may be compromised [32].

## 3. The Strong Families Programme

In response, the United Nations Office on Drugs and Crime (UNODC), implementing a global initiative on prevention and piloting evidence-based family skills prevention in LMIC globally, developed and tested the Strong Families programme. Family-focused interventions were highlighted by the ‘UNODC/WHO International Standards of Drug Use Prevention’ [17]. Substance use prevention strategies have called for a move in the response paradigm; to focus on the people rather than the substances themselves. This modality also fulfils the commitments expressed by the Member States of the United Nations committed to engage in reaching the 17 Sustainable Development Goals (SDGs) on the road to 2030. Family-focused interventions interlink many of the 169 targets of the SDGs. As a result, UNODC has been availing many open-source, low-cost materials for parenting support designed for low- and middle-income countries.

The Strong Families Programme stemmed from the long experience of UNODC inducing such family skills programming in over 40 countries globally. Strong Families is a selective, evidence-informed prevention family skills prevention intervention designed to improve parenting skills, child well-being and family mental health amongst families with children aged between 8 and 15 years. It was tailored specifically for families living in stressful situations, including humanitarian contexts. The programme was designed to be brief and is delivered to groups of families through three 1–2 h training sessions over 3 weeks. It is designed to be resource “light” (requiring an infrastructure that is easy to mobilise and train), evidence-informed, open-source (to facilitate national ownership and scale-up) and cost-effective [33]. The facilitator manual, and associated training, have been prepared to be suitable for programme facilitators with a range of skills, but not necessarily formal training in specific disciplines, with a focus on facilitator interpersonal skills, motivation and access to families.

Since its development in 2017, Strong Families has been piloted in 17 countries across four continents, including, for example, Bangladesh, Afghanistan, Senegal and Lebanon. In our programming efforts, we recognise that low-intensity brief interventions must produce outcomes equivalent to more intensive interventions for at least a proportion of participants [34], such that we have built a thorough evaluation arm alongside our global implementation activities. This has led to Strong Families effectiveness being recognised in a number of published articles, including with Afghan families in Afghanistan [35], with Afghan refugees in Serbia [36], and most recently in an RCT with families in Iran [37]. Strong Families success in supporting families has been recognised by the international community in several ways; it has been mentioned as a tool supporting community-based crime prevention resolution in a UNODC crime commission [38] as an example for programmes for interventions in humanitarian settings under the Helping Adolescents Thrive of WHO and UNICEF for prevention and promotion of Mental Health. We recognise that low-intensity interventions provide the minimum set of skill support to affect the essential skills needed in a broad (universal) context of family profiles. As such, the Strong Families vision of development has not been designed to address all needs of families in such contexts; moreover, it does not necessarily replace any other existing package of family skills available at a national level. It is designed to be used as an introduction to essential family skills to induce the value of such a programmatic approach and encourage assimilation of further intensive or more specialised interventions in the future, based on feasibility. On the other hand, it can be used as a booster social intervention support for any other social family-focused level of intervention available.

Prior to the COVID-19 pandemic, there was already insufficient reach of evidence-based family skills programmes in humanitarian contexts [26,32]. Recognition was already taking shape in the family skills field that technology-based and online formats would be a key path to expanding engagement with families globally [39]. Two years on, remote teleworking technologies are leading an exciting phase of innovation and global reach of services globally, particularly to humanitarian contexts that lack accessibility. Strong Families facilitators are usually trained in the country (face-to-face) over two full days by programme master trainers. As travel restrictions came into place during the onset of COVID-19, while demand for the programme remained high, a Strong Families e-learning platform was developed for facilitator training to meet this need. The platform combines an interactive pdf with multiple videos explaining the most technical exercises, note-taking functions and click and reveal activities to check for comprehension. Participating facilitators also complete assessments and assignments and are requested to submit video recordings of themselves practising activities with fellow trainees. In addition, master trainers lead four 2-h live teaching webinars in which trainees have an opportunity to ask questions and for trainers to provide feedback on the video recordings. To date, the e-learning platform has been made available in 6 languages (three more languages underway), benefitting facilitators from 9 countries, including Cambodia, Bosnia & Herzegovina, Iran, Mexico, Turkmenistan and Zambia. Families have already been reached through facilitators trained by this platform. Data on child mental health, parenting practices and child resilience are currently being assessed.

## 4. Conclusions

In conclusion, despite the highlighted possible reasons for the low uptake of family skills programmes parallel to much evidence highlighting the key role availing such programmes with families living in humanitarian contexts, we hope that such intensive multi-cultural and multi-country settings experience can induce further research and advocate for the value of such family skills-focused work globally. By providing Strong Families as an example of a programme that recognises and addresses the challenges described above, we hope to build on the growing recognition that to extend the reach and impact of family skills interventions, flexible low-intensity interventions are required to meet the direct relevant contextual needs of the target population [28]. Such an experience demonstrates that, despite being a low-intensity intervention, we can significantly affect the stress of such contexts and poor emotional, physical and mental well-being for all family members. We have illustrated that family skills programmes can accommodate the limited infrastructures and changing contexts common to humanitarian contexts when these challenges are first identified and recognised. Through our global programming, we have noted that when family skills programmes are implemented successfully, they are often the first contact adult caregivers might have with professional service providers. Accordingly, such interventions can also carry a significant public health implication of expanding their intended goal to serve as a natural entry point for identifying what other challenges family members may be experiencing and/or linking them to as access or gateway points to any other available social or health service provisions at the community level. This might be for health issues, domestic violence, sexual reproductive health, or socioeconomic welfare. Moving forward, more research is needed to unpack the content, delivery mechanisms and reach of family skills programmes to further aid programme developers in investing in efforts that might provide significant sustained impact for families in humanitarian contexts.

## Data Availability

Not applicable.

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
