# Peer review of "Bridging the Gap between the Pressing Need for Family Skills Programmes in Humanitarian Settings and Implementation"

_ijerph, 2022, doi:10.3390/ijerph19042181_

Round 1
Reviewer 1 Report
Thank you for providing the opportunity to review this paper. The manuscript describes barriers successful implementation of family skills interventions in humanitarian crisis contexts. Additionally, the manuscript describes the Strong Families program, and reviews initial evaluation data about the program.
Overall this is an important paper for practitioners in the field and organizations designing interventions.
My specific comments are as follows:
Structure: Consider addition sections of the manuscript. (ie there is "introduction;" consider "review of existing programs". "barriers to implementation," "strong families," "conclusions," or other headings to guide the reader and provide ease of reference for review later in the field.
Lines 51-55: I would consider reframing this statement, to match the philosophy a little more of the Strong Families program. This statement places the burden on stressed parents and caregivers rather than the environment in which the caregivers are caring for children.
Lines 75-77: If space allows, consider briefly discussing context and outcomes of these studies.
Lines 161-170: This section begins with a discussion of substance use prevention. It may be helpful to introduce substance use prevention earlier in the paper, or here to focus on the Strong Families program itself and the use in humanitarian crises specifically. Perhaps starting with a description of the program, including its philosophical framework, or briefly the aspects that help to set it apart from prior programs, could be helpful. Once the program has been introduced the history of use by UNODC could be described in more detail.
Lines 246-255: I think you could even make a stronger argument here for the vital importance of family skills training programs that are suitable for humanitarian settings, in addition to your call for further research.
Author Response
Reviewer 1
Thank you for providing the opportunity to review this paper. The manuscript describes barriers successful implementation of family skills interventions in humanitarian crisis contexts. Additionally, the manuscript describes the Strong Families program, and reviews initial evaluation data about the program. Overall, this is an important paper for practitioners in the field and organizations designing interventions.
We thank the reviewer for their comments for their appreciation that this piece will make a valuable addition to the field.
My specific comments are as follows:
Structure: Consider addition sections of the manuscript. (ie there is "introduction;" consider "review of existing programs". "barriers to implementation," "strong families," "conclusions," or other headings to guide the reader and provide ease of reference for review later in the field.
Additional headings have been added such as ‘The Strong Families Programme’, ‘Barriers to implementation’, and ‘Conclusion’,
Lines 51-55: I would consider reframing this statement, to match the philosophy a little more of the Strong Families program. This statement places the burden on stressed parents and caregivers rather than the environment in which the caregivers are caring for children.
We appreciate this note and have edited the statement as guided.
Lines 75-77: If space allows, consider briefly discussing context and outcomes of these studies.
Further information has been provided on the studies, though due to the already increased word count there is no scope to go into specific details of each of the 4 referenced studies.
Lines 161-170: This section begins with a discussion of substance use prevention. It may be helpful to introduce substance use prevention earlier in the paper, or here to focus on the Strong Families program itself and the use in humanitarian crises specifically. Perhaps starting with a description of the program, including its philosophical framework, or briefly the aspects that help to set it apart from prior programs, could be helpful. Once the program has been introduced the history of use by UNODC could be described in more detail.
Rather than restructure this section, we have taken the reviewers useful comment and have now threaded through earlier on in the introduction drug use as a risk factor of experiencing stress and the key role effective caregiving can place in reducing this risk. We believe this coherently and logical introduces the UNODC Standards on drug use prevention as a good opening for the following section.
Lines 246-255: I think you could even make a stronger argument here for the vital importance of family skills training programs that are suitable for humanitarian settings, in addition to your call for further research.
We thank the reviewer for this comment. We have now expanded the conclusion further, while still trying to not expand the word count too much over the guidelines for such a short communication

Reviewer 2 Report
The communication shows the effectiveness of the Strong Families program in preventing mental health problems in children and adolescents living in humanitarian contexts (1/5% of the world population).
A relevant contribution is the online training of program facilitators in response to the Covid epidemic. It also includes references on the cross-cultural effectiveness of the program. It also points out the difficulties of its implementation, both because of the need to address basic needs first, as well as the lack of rigorous controls in its implementation.
It highlights the recognition of the need for a PARADIGM CHANGE that involves focusing on people (Alonso-Stuyck, P. Which Parenting Style Aliens Healthy Lifestyles in Teenage Children? Proposal for a Model of Integrative Parenting Styles. Int. J. Environ. Res. Public Health 2019, 16, 2057), are tailored to their needs, rather than content. Compassion-based programs move along these lines (Kirby, JN Nurturing Family Environments for Children: Compassion-Focused Parenting as a Form of Parenting Intervention. Sci. Educ. 2020, 10, 3)?
Some lines of hope could be included with Boris Cyrulnik's contributions on resilience https://www.torrossa.com/en/resources/an/2468928 having grown up in these contexts, and in the systemic generalization of the benefits of the program to both family and humanitarian contexts.
Translated with www.DeepL.com/Translator (free version)
La comunicación muestra la efectividad del programa Familias Fuertes para prevenir problemas de salud mental en niños y adolescentes que viven en contextos humanitarios (1/5% de la población mundial).
Un aporte relevante es la capacitación en línea de los facilitadores del programa en respuesta a la epidemia de Covid. también incluye referencias sobre la eficacia intercultural del programa. Y también señala las dificultades de su implementación, tanto por la necesidad de atender primero las necesidades básicas, como por la falta de controles rigurosos en su implementación.
Destaca el reconocimiento de la necesidad de un CAMBIO DE PARADIGMA que implica centrarse en las personas (Alonso-Stuyck, P. Which Parenting Style Aliens Healthy Lifestyles in Teenage Children? Proposal for a Model of Integrative Parenting Styles. Int. J. Environ. Res. Public Health 2019, 16, 2057), se adaptan a sus necesidades, más que al contenido. Los programas basados ​​en la compasión se mueven en esta línea (Kirby, JN Nurturing Family Environments for Children: Compassion-Focused Parenting as a Form of Parenting Intervention. Sci. Educ. 2020, 10, 3)?
Algunas líneas de esperanza podrían incluirse con las aportaciones sobre la resiliencia de Boris Cyrulnik https://www.torrossa.com/en/resources/an/2468928 haber crecido en estos contextos, y en la generalización sistémica de los beneficios de la programa tanto para familiares como para contextos humanitarios.
Author Response
Reviewer 2
The communication shows the effectiveness of the Strong Families program in preventing mental health problems in children and adolescents living in humanitarian contexts (1/5% of the world population). A relevant contribution is the online training of program facilitators in response to the Covid epidemic. It also includes references on the cross-cultural effectiveness of the program. It also points out the difficulties of its implementation, both because of the need to address basic needs first, as well as the lack of rigorous controls in its implementation.
It highlights the recognition of the need for a PARADIGM CHANGE that involves focusing on people (Alonso-Stuyck, P. Which Parenting Style Aliens Healthy Lifestyles in Teenage Children? Proposal for a Model of Integrative Parenting Styles. Int. J. Environ. Res. Public Health 2019, 16, 2057), are tailored to their needs, rather than content. Compassion-based programs move along these lines (Kirby, JN Nurturing Family Environments for Children: Compassion-Focused Parenting as a Form of Parenting Intervention. Sci. Educ. 2020, 10, 3)?
Some lines of hope could be included with Boris Cyrulnik's contributions on resilience https://www.torrossa.com/en/resources/an/2468928 having grown up in these contexts, and in the systemic generalization of the benefits of the program to both family and humanitarian contexts.
We thank the reviewer for this comment. We have threaded into the introduction a statement on hope and resilience as a key protective factor for children hat results from having supportive caregivers.

Reviewer 3 Report
I think there is a need for some better transitions and connecting the dots for readers.
I think we also get to the end of this paper, that is well read, and are expecting more.
Although it is labeled as a short communication, and therefore there is not going to be a study attached, there's little "so what" to the manuscript. What's the point? What do we do now? What do we do with this article? What are some tangible next steps? I think you begin to do this, but then is cut short.
Author Response
Reviewer 3
I think there is a need for some better transitions and connecting the dots for readers.
This short communication has been reviewed to ensure smother transitions between sections. Also, in some cases headings have been added for a more logical flow.
I think we also get to the end of this paper, that is well read, and are expecting more.
We thank the reviewer for this comment. We have now expanded the conclusion further, while still trying to not expand the word count too much over the guidelines for such a short communication.
Although it is labeled as a short communication, and therefore there is not going to be a study attached, there's little "so what" to the manuscript. What's the point? What do we do now? What do we do with this article? What are some tangible next steps? I think you begin to do this, but then is cut short.
We have rewritten the conclusion to address this comment and provide a clearer overview of what next steps are recommended, including more research to unpack the content, delivery mechanisms and reach of family skills programmes to further aid programme developers in investing in efforts that might provide significant sustained impact for families in humanitarian contexts. We have also better highlighted the aim and need for such a communication in this field.
